# Synergistic Effects of Probiotic and Omega-3 Supplementation with Ultra-Short Race Pace Training on Sprint Swimming Performance

**DOI:** 10.3390/nu17142296

**Published:** 2025-07-11

**Authors:** Ideh Maymandinejad, Mohammad Hemmatinafar, Ralf Jäger, Babak Imanian, Maryam Koushkie Jahromi, Katsuhiko Suzuki

**Affiliations:** 1Department of Sport Science, Faculty of Education and Psychology, Shiraz University, Shiraz 71345, Iran; 2Increnovo LLC, Whitefish Bay, WI 53217, USA; 3Faculty of Sport Sciences, Waseda University, 2-579-15 Mikajima, Tokorozawa 359-1192, Japan

**Keywords:** sprint swimming performance, probiotic supplementation, omega-3 fatty acids, ultra-short race pace training (USRPT), synbiotic effects

## Abstract

**Background**: Optimal nutrition and training regimens are essential for athletes to maximize performance and recovery. Probiotic supplementation, through the modulation of the gut microbiota, and omega-3 fatty acids, known for their anti-inflammatory properties, may enhance physiological adaptations when combined with targeted training. This study evaluated the effects of probiotics and omega-3 supplementation, alongside ultra-short race pace training (USRPT), on performance metrics in competitive sprint swimmers. **Methods**: In this double-blind, placebo-controlled study, 60 male sprint swimmers (age: 19.2 ± 3.6 years; height: 182.2 ± 5.2 cm; weight: 81.6 ± 4.4 kg) with a minimum of five years of training experience, were randomly assigned to six groups (n = 10 per group): (1) Control (CON), (2) USRPT only, (3) Placebo + USRPT (PLA + USRPT), (4) Probiotics + USRPT (PRO + USRPT), (5) Omega-3 + USRPT (OMEGA + USRPT), and (6) Probiotics + Omega-3 + USRPT (PRO + OMEGA + USRPT). Over the eight-week intervention, the participants in PRO + USRPT consumed one multi-strain probiotic capsule daily (4.5 × 10^11^ CFU) and a placebo capsule. Those in OMEGA + USRPT ingested 1000 mg of fish oil after lunch (500 mg EPA and 180 mg DHA per capsule) paired with a placebo capsule. The combined supplementation group (PRO + OMEGA + USRPT) received both probiotic and omega-3 capsules. The PLA + USRPT group consumed two starch capsules daily. The USRPT protocol was implemented across all the training groups, where the swimmers performed 17 sets of 25 m and 12.5 m sprints based on weekly recorded race times. Performance assessments included pre- and post-test measurements of sprint times (50 m and 100 m freestyle), vertical jump tests (both in water and on dry land), and other strength and endurance metrics (reaction time, agility *T*-test, sprint index, fatigue index, and velocity). **Results**: The combined intervention of probiotics and omega-3 with USRPT produced the greatest improvements in performance. The PRO + OMEGA + USRPT group reduced 50 m freestyle time by 1.92% (*p* = 0.002, pEta^2^ = 0.286) and 100 m freestyle time by 2.48% (*p* = 0.041, pEta^2^ = 0.229), demonstrating significant Time × Group interactions consistent with a synergistic effect. Additionally, the sprint index improved (pEta^2^ = 0.139, *p* = 0.013) and reaction time decreased (pEta^2^ = 0.241, *p* = 0.009) in the combined group, indicating enhanced anaerobic capacity and neuromuscular responsiveness compared to single interventions. **Conclusions**: This study suggests that combining probiotics and omega-3 supplementation with USRPT leads to synergistic improvements in sprint swimming performance, enhancing anaerobic power and recovery beyond what is achieved with individual interventions. This integrated approach may provide a practical strategy for competitive swimmers seeking to optimize their performance. Future studies should incorporate mechanistic markers, longer intervention durations, and diverse athlete populations to clarify further and extend these findings.

## 1. Introduction

Swimming is one of the world’s most popular sports and the second-largest sport based on the number of athletes at the Olympic Games [1]. Swimmers require a combination of physical abilities, such as strength, speed, and endurance, to reach peak performance [2]. To enhance swimmers’ physical abilities and boost their performance, it is essential to include an appropriate training program and a sport-specific, balanced diet [2,3]. This allows athletes to achieve their peak performance levels while also aiding in speedy recovery and reducing the risk of injuries [4]. Many athletes incorporate nutritional supplements into their diet to further enhance their training objectives. These supplements cater to various needs like addressing nutrient deficiencies, improving recovery, facilitating muscle synthesis, increasing energy levels, and enhancing performance in specific sports or activities [4]. An example of this is sprint swimming, which is a high-intensity sport of short duration that relies more on anaerobic metabolism for energy [5]. Due to the nature of their activity, sprint swimmers have a robust buffer system, a high lactate threshold, an efficient anaerobic energy system, and the ability to produce energy quickly [5]. Additionally, sprint swimmers must enhance muscle mass, strength, power, reaction time, agility, explosive power, sprint performance, and endurance to excel during training and competitions, all of which can be impacted by a combination of exercise regimen, dietary strategies, and supplementation [6].

Interest has recently grown in manipulating gut health as a strategy to improve performance [7]. The human gut microbiota, comprising over 10^14^ microorganisms, plays a central role in nutrient absorption and immune regulation [7,8]. Probiotics—live microorganisms that confer health benefits when consumed in adequate amounts—can modulate intestinal permeability, improve nutrient uptake, and reduce local inflammation, thereby supporting overall homeostasis [7,8]. Certain strains, such as *Lactiplantibacillus plantarum*, have been shown to enhance gut barrier function [9], facilitating the absorption of minerals, peptides, and amino acids, while also supporting villus growth [10]. Probiotics can strengthen immune responses by improving antigen presentation and activating T and B lymphocytes [11], which is especially important for athletes under heavy training loads. They may also reduce muscle catabolic pathways linked to NF-kB activity [11,12], and some studies suggest probiotics can extend time to fatigue by modulating tryptophan metabolism [7]. Additionally, multi-strain probiotic supplementation has been shown to increase time to exhaustion in athletes [12], and probiotics may influence lactate handling, potentially aiding in its clearance or utilization by lactate-consuming bacteria, though direct evidence in athletes is still limited [7,13]. Thus, probiotics could support anaerobic capacity and performance in swimmers [7].

Omega-3 fatty acids, particularly eicosapentaenoic acid (EPA) and docosahexaenoic acid (DHA), are incorporated into cell membrane phospholipids and act as precursors to anti-inflammatory mediators [14,15]. They contribute to the health of the heart, skeletal muscle, and brain [16], and have been implicated in improved nutrient responsiveness and exercise [17]. Mechanistically, omega-3s can enhance amino acid uptake, stimulate the AKT/mTOR pathway, promote muscle protein synthesis, and inhibit catabolic processes involving MAFbx, MuRF, NF-kB, and oxidative stress [18]. They also reduce inflammation by inhibiting COX-2 and shifting eicosanoid production toward less pro-inflammatory profiles [14]. However, research on omega-3 supplementation alone in exercise settings is mixed, with some studies showing modest or unclear effects on recovery and performance [19,20]. Notably, omega-3 intake has been linked to increased gut microbial diversity [21], suggesting possible indirect prebiotic-like interactions [22]. Recent expert consensus from the International Scientific Association for Probiotics and Prebiotics (ISAPP) expanded the definition of prebiotics to include non-carbohydrate compounds such as polyunsaturated fatty acids, supporting the plausibility of omega-3s contributing to a synbiotic-like interaction [22]. This perspective raises the hypothesis that combining probiotics with omega-3 fatty acids might yield synergistic or complementary benefits for athletes, paralleling the concept of synbiotics traditionally employed to enhance gut health. Indeed, a synbiotic approach—co-administering prebiotics with probiotics—has been shown to modulate gut microbiota composition more effectively than probiotics alone [22]. Although omega-3s are not classical prebiotics, their intake has been associated with increased gut microbial diversity and shifts toward beneficial taxa, suggesting potential prebiotic-like properties. Thus, the co-supplementation of omega-3 fatty acids and probiotics could, in theory, result in additive or synergistic effects on gut integrity, immune modulation, and systemic inflammation, ultimately enhancing physiological adaptations and athletic performance when combined with structured exercise training.

Swimmers require a specialized training program to maximize their performance. Training remains foundational to sprint swimming performance. Ultra-short race pace training (USRPT) involves breaking race distances into shorter intervals performed at race pace, which effectively targets the physiological demands of competition [23]. USRPT can reduce heart rate, perceived exertion, and blood lactate while improving speed, VO_2_ max, strength, anaerobic capacity, and overall performance in sprint swimmers [24,25]. The assessments of such training encompass a range of indicators from explosive power and muscular endurance to agility and sprint-specific indices [26].

Given the physiological demands of sprint swimming and the pivotal role of anaerobic power, optimizing performance through targeted interventions is essential. Probiotic supplementation, known for modulating the gut microbiota and enhancing nutrient absorption, alongside the anti-inflammatory actions of omega-3 fatty acids, offers a promising strategy to support athletic performance [7,14]. By improving gut health, reducing systemic inflammation, and potentially aiding recovery, these interventions may help athletes better tolerate the high-intensity demands of USRPT, minimizing fatigue and overtraining risks. Moreover, USRPT itself aligns training intensity closely with competitive needs [27], complementing these nutritional approaches. Despite growing interest, few studies have examined whether combining probiotics and omega-3s produces a synergistic or synbiotic-like effect that enhances sprint swimming performance beyond individual supplementation [28]. Although each of these interventions has individually demonstrated potential benefits, no prior research has investigated the combined effects of probiotics and omega-3 supplementation alongside a USRPT regimen in competitive sprint swimmers. Therefore, this study aimed to address this gap by examining whether integrating these nutritional strategies with USRPT could synergistically enhance sprint swimming performance, anaerobic power, and recovery indicators. Addressing this question is the primary objective of the current study. Finally, only male swimmers were recruited to control for hormonal fluctuations related to the menstrual cycle, which could impact performance, metabolism, and recovery.

## 2. Methodology

### 2.1. Participants

The current study involved 60 male sprint swimmers with at least five years of competitive experience, including six weekly training sessions and a history of participation in championships. The anthropometric characteristics of the participants are presented in Table 1. Study procedures were thoroughly explained to the participants, and written informed consent, along with completed Physical Activity Readiness Questionnaire (PAR-Q) forms, were obtained. The participants were screened to ensure they had no history of disease or known medical conditions, no allergies to probiotics or omega-3 supplements, and were not taking performance-enhancing drugs during the intervention period. Additionally, the participants refrained from smoking, consuming alcohol, or drinking caffeinated beverages for 24 h before data collection. All 60 participants were professional swimmers who volunteered for the study and met the outlined inclusion criteria. The study protocol was reviewed and approved by the Research Ethics Committee of the Faculty of Psychology and Educational Sciences, Shiraz University, Shiraz, Iran (Ethics Approval Code: IR.US.PSYEDU.REC.1403.045, Approval Date: 10 July 2024), and was conducted in full compliance with the Declaration of Helsinki.

### 2.2. Sample Size Calculation

The required sample size was calculated using the repeated measures design equation integrated into the G*Power software (version 3.1.9.7). The calculation was based on an effect size of 0.25, a power of 80%, a type I error probability of 0.05, six groups, and two repeated measurements [29]. The analysis determined that a minimum of 55 participants would be required to ensure adequate statistical power for the exercise, omega-3, and probiotics intervention. To account for an estimated dropout rate of approximately 10%, the final aim was to recruit 60 eligible participants.

### 2.3. Study Design

This study used a randomized, double-blind, placebo-controlled design (Figure 1). Initially, the participants attended a session where a general practitioner evaluated their overall health. During this session, the participants provided written informed consent after a detailed explanation of the study protocol, benefits, risks, and potential side effects. They also completed the PAR-Q and a food registration questionnaire to assess dietary and supplement intake that could influence the gut microbiota, including fermented foods (e.g., yogurt, kefir, and kimchi), and supplements such as probiotics, prebiotics, and omega-3 fatty acids. Additionally, the participants underwent a familiarization session with the equipment, testing techniques, and Shiraz University’s swimming pool. The participants were instructed to avoid consuming caffeinated products and perform vigorous exercise at least 24 h before the experiment to control for acute effects on measured variables. The pre-test phase occurred over two consecutive days between 9:00 AM and 12:00 PM. On the first day, after warming up, the participants performed dry-land tests, including the vertical jump height (VJH), repeated vertical jump (RVJ), reaction time (RT), medicine ball throw for upper-body explosive power (TM), body fat percentage (FP), muscle mass percentage (MP), and agility *T*-test. On the second day, the participants completed in-water tests, including 50 m freestyle (50 m fr), 100 m freestyle (100 m fr), in-water vertical jump height (WVJH), repeated water vertical jump in 60 s (RWVJ), start distance (SD), push-off distance (PD), sprint index (SI), fatigue index (FI), and velocity (V). There were approximately ten minutes of active recovery between each test. The participants were then randomly assigned to six groups (n = 10 per group): (1) Control (CON), (2) USRPT only, (3) Placebo + USRPT (PLA + USRPT), (4) Probiotics + USRPT (PRO + USRPT), (5) Omega-3 + USRPT (OMEGA + USRPT), and (6) Probiotics + Omega-3 + USRPT (PRO + OMEGA + USRPT). Each group received its assigned supplement along with instructions for its proper use. The participants were also advised to maintain their usual diet throughout the intervention period. To standardize pre-exercise conditions, all the participants consumed the same breakfast, containing 250 kcal (45 g carbohydrates, 9 g protein, and 5 g fat) one hour and thirty minutes before each exercise session [30]. The post-test was conducted eight weeks after the pre-test at the same time (9:00 AM to 12:00 PM) under similar environmental conditions (Figure 2). During both test sessions, the participants were allowed to drink water ad libitum. All the participants were members of the same training team, followed the same training program, and were supervised by qualified trainers throughout the study.

### 2.4. Blinding Procedure

To ensure methodological rigor and minimize bias, the study utilized the Sequentially Numbered, Opaque, Sealed Envelope (SNOSE) method for group allocation. An independent researcher not involved in the study prepared identical, opaque, and sequentially numbered envelopes, each containing a pre-determined group allocation code. The participants were assigned an envelope in the order of their enrollment, and each envelope was opened only after the pre-test phase was completed, ensuring that neither the participants nor the researchers were aware of group assignments at the start of the intervention. Furthermore, supplements were prepared and coded by a separate third-party researcher using identical packaging to maintain the double-blind design. This robust procedure aligns with established practices to reduce potential biases and enhance the validity of results [31].

### 2.5. Training Protocol

In addition to their standardized training regimen, which applied to all the participants, the swimmers engaged in the ultra-short race pace training (USRPT) protocol over a duration of eight weeks, with three sessions per week. At the end of each week, the swimmers’ 100 m and 50 m freestyle times were recorded, and their times were divided by four to determine the target pace for the USRPT sessions. Swimming times were recorded using an electronic stopwatch (Stopwatch Selecta, W10710, Waterfly, Shinjuku City, Tokyo) by two experienced coaches from the research team to ensure accuracy. Based on these times, the swimmers performed 17 repetitions of 25 m and 17 repetitions of 12.5 m at their calculated race pace. Each 25 m repetition was followed by a 10 s rest period, while 12.5 m repetitions were followed by a 5 s rest period. A 5 min active rest was also provided between the 25 m and 12.5 m sets. The participants were instructed to maintain a pace consistent with their 100 m and 50 m event times throughout the protocol. To progressively increase training intensity, two repetitions were added to the protocol each week [23,27,32].

### 2.6. Supplementation Protocol

The participants were assigned to specific supplementation regimens over eight weeks. The PRO + USRPT group took one daily probiotic capsule (Comflor PRO, Farabiotic Company, Tehran, Iran) containing eight bacterial strains and a total dose of 4.5 × 10^11^ CFU (Table 2), along with one placebo capsule [12,33]. The OMEGA + USRPT group consumed one fish oil capsule (EuRho Vital, Bönen, Germany) daily, providing 1000 mg of fish oil, 500 mg of EPA, and 180 mg of DHA, along with one placebo capsule. The PRO + OMEGA + USRPT group combined both supplements, consuming one probiotic capsule during lunch and one omega-3 capsule after lunch. The placebo group (PLA + USRPT) received two starch capsules, carefully designed to match the appearance, color, and size of the probiotic and fish oil capsules, thereby maintaining blinding. The participants received EuRho Vital omega-3 fish-oil capsules, specifically formulated to be odorless, thereby minimizing the risk of unblinding due to sensory detection. The participants were not informed of their group allocation, and precautions were taken to prevent them from observing the supplements provided to other participants. This approach ensured that all the participants consumed two capsules daily, maintaining the integrity of the double-blind design (Figure 1). Participant compliance with supplementation exceeded 95%, as verified by weekly pill counts and individual daily intake logs maintained over the intervention. This high adherence (>95%) is consistent with or exceeds compliance levels typically reported in comparable exercise-nutrition interventions [7].

### 2.7. Functional Tests

#### 2.7.1. The 50 m and 100 m Freestyle

All the swimmers followed a standardized warm-up routine, which included a 600 m crawl at a moderate pace and three 50 m crawl trials with progressively increasing speed. The main task involved completing a 50 m swim (one lap of a 50 m pool) and a 100 m swim (two laps) as quickly as possible. Upon hearing the signal “ready, start,” they began swimming, and their time was recorded using a stopwatch. Timing stopped when the swimmer’s hand touched the wall after the respective distance [34,35]. The tests were conducted under consistent conditions in an indoor swimming pool measuring 50 m in length, 25 m in width, and a depth ranging from 1 to 4 m. The water temperature was maintained at 26 °C.

#### 2.7.2. Start Distance (SD)

The start distance (SD) test was conducted to evaluate the maximum distance swimmers could achieve solely through the propulsion generated during the start without using swimming strokes or fly kicks. This test is commonly used to measure the effectiveness of the initial propulsion phase in competitive swimming. Studies such as those by Matúš and Kandráč (2020) [36] have utilized kinematic analysis to measure parameters like flight distance, glide distance, and maximum depth during the underwater phase of the kick start. Similarly, Barlow et al. (2014) [37] demonstrated the influence of start positions on swimming performance by analyzing the distance covered up to 15 m, emphasizing the importance of optimized start mechanics. These findings highlight the relevance of measuring start distance in assessing initial propulsion efficiency and its impact on overall swimming performance [36,37].

#### 2.7.3. Push-Off Distance (PD)

The push-off distance (PD) test measured the maximum distance swimmers could achieve solely through the propulsion generated by push-off in the water, without engaging in swimming strokes or kicking. This test is a standard method for evaluating the effectiveness of underwater push-off mechanics and their contribution to overall swimming performance. Studies, such as those by Matúš and Kandráč (2020) [36], have used detailed kinematic analysis to examine parameters like glide distance and underwater movement efficiency during the push-off phase. Similarly, Barlow et al. (2014) [37] analyzed the distance covered during underwater glides following a push-off and highlighted the importance of optimizing push-off mechanics to improve performance in competitive swimming. These findings highlight the significance of the push-off distance test in assessing underwater propulsion and its impact on performance [36,37].

#### 2.7.4. In-Water Vertical Jump Test

The in-water vertical jump (WVJ) test measured explosive leg power and assessed vertical propulsion capabilities in a submerged environment. This test followed the methodology described by Platanou (2006) [38]. The participants performed the test in a pool with a water depth of approximately 1 m, ensuring that swimmers could maintain a stationary position without their feet leaving the pool floor during the jump. A vertical board, marked with a centimeter scale, was installed on the pool wall, and a video camera placed perpendicular to the board recorded the trials to ensure accurate measurement. Each swimmer performed three vertical jumps, starting from a stationary position without external momentum or the use of arm thrust. The highest point reached during each jump was recorded, and the best result from the three trials was used for analysis. This test is a reliable and valid method for evaluating underwater explosive power, providing valuable insight into the participants’ lower-body strength and propulsion efficiency in aquatic conditions [38].

#### 2.7.5. Repeated Vertical Jump Test in Water (RWVJ)

To evaluate muscular endurance in power, swimmers performed repeated vertical jumps in a 1 m deep pool for 60 s. The best vertical jump height from the initial in-water vertical jump test was recorded as the target height. Swimmers were instructed to continuously perform vertical jumps, aiming to reach the target height with each attempt. The total number of successful repetitions (i.e., those reaching the target height) within the 60 s duration was recorded. This protocol, adapted from Platanou (2006) and Stemm & Jacobson (2007), provides a reliable measure of endurance in explosive movements specific to aquatic environments [38,39].

#### 2.7.6. Anaerobic Capacity Test

The anaerobic capacity test consisted of six consecutive all-out 15 m sprints, each initiated from a wall push-off. After each sprint, the swimmers continued to swim at a low intensity until reaching the 25 m mark of the pool, allowing for active recovery. This 15 m sprint + 10 m active swim recovery formed one repetition within the 50 m pool. This setup ensured that all sprints were performed under consistent, controlled conditions within a standard pool environment. The test protocol, adapted from the established literature [40], was designed to evaluate anaerobic power, velocity, and fatigue resistance using the following metrics:

Sprint Index (SI): (Weight × Distance^2^) ÷ Time^3^

This metric reflects the swimmer’s power output during each sprint.

Velocity (V): Distance ÷ Time

Velocity measures the average speed over the 15 m sprint distance.

Fatigue Index (FI): ((Max SI − Min SI) ÷ Total Time) × 100

FI evaluates the percentage decrease in performance across the sprints, indicating the swimmer’s ability to sustain high-intensity effort over time [26]. This testing approach combines these metrics to comprehensively assess anaerobic performance, capturing peak power output and fatigue resistance.

#### 2.7.7. Reaction Time (RT)

The reaction time (RT) test evaluated the swimmers’ ability to respond to auditory stimuli. Following the methodologies described in previous studies, the swimmers were instructed to perform a vertical jump as quickly as possible upon hearing a predefined sound signal [41]. Reaction time, defined as the interval between the auditory stimulus and the initiation of the jump, was measured using a DSI device (Danesh Salar Company, Tehran, Iran). The DSI device provided precise and reliable measurements, ensuring the accuracy of recorded reaction times [42]. This test offers a robust assessment of athletes’ neuromuscular responsiveness and reaction speed under controlled conditions, in line with established auditory reaction time measurement methodologies.

#### 2.7.8. Agility Test (T-Test)

The agility test was conducted using the T-Test protocol to assess the participants’ speed, agility, and ability to change direction. The setup included four cones (A, B, C, and D) arranged in a “T” formation. The participants began at cone A. Upon the timer’s command, the participant sprinted to cone B and touched its base with their right hand. They then turned left and shuffled sideways to cone C, touching its base with their left hand. Next, they shuffled sideways to the right to cone D, touching its base with their right hand. Finally, they shuffled back to cone B, touching it with their left hand, and ran backward to cone A. The stopwatch was stopped as the participant crossed cone A [43] (Figure 3).

#### 2.7.9. Medicine Ball Throw Test

The medicine ball throw (TM) test evaluated the explosive strength of the participant’s upper body muscles. The participants stood at a designated starting line with their feet slightly apart, facing the direction of the throw. While holding the medicine ball, their hands were positioned on the sides and slightly behind the center of the ball. The throwing motion mimicked a soccer/football throw-in, with the ball pulled back behind the head and then propelled forward with maximum force. The participants were allowed to step across the line during or after the throw to maximize their effort. Each participant performed three throws, and the distance from the starting line to the point where the ball first landed was measured. The best result from the three attempts was recorded as the final score. This test is a validated and widely accepted measure of upper body explosive power, providing reliable performance indicators across various athletic disciplines (Mayhew et al., 2005) [44].

#### 2.7.10. Body Composition Analysis

Body composition, a key determinant of athletic performance, was assessed using the Accuniq BC380, a bioelectrical impedance analyzer manufactured by SELVAS Healthcare in Korea. This device measures body fat percentage (FP) and muscle mass percentage (MP) using advanced multi-frequency bioelectrical impedance analysis (BIA), which has been validated for accuracy and reliability in various populations, including athletes. Studies show strong correlations between Accuniq devices and dual-energy X-ray absorption (DXA), supporting its use for precise and non-invasive body composition analysis [45].

#### 2.7.11. Sargent’s Jump Test

The Sargent Jump Test measured the participants’ vertical jump height (VJH) and explosive leg power. The participants chalked the tips of their fingers and stood next to a wall at the shortest possible distance, keeping both feet flat on the ground. They reached upward with one hand to make the highest possible mark on the wall (M1). The participants then jumped as high as possible from a static position and marked the wall again at the highest point of their jump (M2). The vertical jump height was calculated as the difference between M2 and M1. Each participant performed three attempts, with a one-minute rest between trials, and the best score was recorded as the final result. This test is a validated and reliable method for assessing explosive lower-body power and is commonly used in athletic performance evaluations [46].

The participants performed repeated vertical jumps (RVJs) for 60 s to further assess muscular endurance in power. The best vertical jump height from the Sargent Jump Test was set as the target height. For each attempt, the participants were instructed to perform continuous vertical jumps, aiming to reach the target height. The total number of successful repetitions within the 60 s period was recorded. This protocol provides a reliable measure of muscular endurance in explosive movements [47].

### 2.8. Statistical Analyses

All data were analyzed using descriptive and inferential statistical methods. The data distribution was tested for normality using the Kolmogorov–Smirnov test. The mixed repeated measure analysis ANOVA test was employed to identify the main effects, and the Bonferroni post hoc test was used to determine pairwise differences by adding different Syntax codes. Additionally, repeated-measures ANOVA models included Time × Group interaction terms to formally assess whether performance changes over time differed between intervention groups, providing statistical evidence for potential synergy. Partial eta squared (pEta^2^) was calculated to estimate effect size, with thresholds of 0.01 (small), 0.06 (medium), and 0.14 (large) interpreted based on conventional guidelines for η^2^ magnitude [48]. We designed the study and calculated the sample size based on our main (primary) outcomes. We also explored other outcomes without strict multiple-testing corrections, treating them as exploratory, to generate ideas for future studies. Data are presented as mean ± standard deviation (SD), and a significance level of *p* ≤ 0.05 was set for all the analyses. The SPSS software (version 26, IBM-SPSS Inc., Chicago, IL, USA) was used for all the statistical analyses. Additionally, figures were generated using GraphPad Prism (version 9.0.0, GraphPad Software, San Diego, CA, USA) to visually present the results.

## 3. Results

Table 1 presents descriptive statistics for the participants’ anthropometric characteristics. Table 3 summarizes the means and standard deviations of the measured variables, including in-water (50 m freestyle, 100 m freestyle, WVJH, RWVJ, SD, PD, SI, FI, and V) and dry-land (VJH, RVJ, RT, TM, FP, MP, and agility) performance metrics.

### 3.1. In-Water Performance Metrics

The 50 m Freestyle (50 m fr): The results of the repeated measure analysis test showed that the main effect of the intervention was significant on 50 m fr (F_1.00_ = 48.428, *p* = 0.001, pEta^2^ = 0.473). Also, the interaction effect of time and intervention (Time × Group) was significant (F_5.00_ = 4.328, *p* = 0.002, pEta^2^ = 0.286). In addition, the results of the Bonferroni test showed that in the USRPT (*p* = 0.033), PLA + USRPT (*p* = 0.017), OMEGA + USRPT (*p* = 0.002), PRO + USRPT (*p* = 0.001), and PRO + OMEGA + USRPT (*p* = 0.001) groups, 50 m fr records significantly decreased in the post-test compared to the pre-test. However, there was no significant difference in the pre-test and post-test 50 m fr records in the CON group (*p* > 0.05). Additionally, no significant differences were observed between the research groups in the pre-test and post-test 50 m fr records (*p* > 0.05; Figure 4, Table 4).

The 100 m Freestyle (100 m fr): The results of the repeated measure analysis test showed that the main effect of the intervention was significant on 100 m fr (F_1.00_ = 54.192, *p* = 0.001, pEta^2^ = 0.501). Additionally, the interaction effect of time and intervention (Time × Group) was significant (F_5.00_ = 1.598, *p* = 0.041, pEta^2^ = 0.229). The results of the Bonferroni test indicated substantial decreases in 100 m fr times in USRPT (*p* = 0.043), PLA + USRPT (*p* = 0.024), OMEGA + USRPT (*p* = 0.001), PRO + USRPT (*p* = 0.002), and PRO + OMEGA + USRPT (*p* = 0.001) groups in the post-test compared to the pre-test. However, no significant difference was found between the pre-test and post-test 100 m fr times in the CON group (*p* > 0.05). Additionally, there were no significant differences between the research groups in the pre-test and post-test 100 m fr records (*p* > 0.05; Figure 4, Table 4).

In-Water Vertical Jump Height (WVJH): The results of the repeated measure analysis test showed that the main effect of the intervention was significant on WVJH (F_1.00_ = 29.042, *p* = 0.001, pEta^2^ = 0.350). However, the interaction effect of time and intervention (Time × Group) was not significant (F_5.00_ = 0.823, *p* = 0.539, pEta^2^ = 0.071). The Bonferroni test results indicated significant increases in WVJH in the OMEGA + USRPT (*p* = 0.002) and PRO + OMEGA + USRPT (*p* = 0.001) groups in the post-test compared to the pre-test. However, no significant improvement was observed in WVJH in the PRO + USRPT, USRPT, PLA + USRPT, or CON groups (*p* > 0.05). Additionally, there were no significant differences between the research groups in the pre-test and post-test WVJH records (*p* > 0.05; Figure 4, Table 4).

Repeated Vertical Jumps in Water (RWVJs): The results of the repeated measure analysis test showed that the main effect of the intervention was significant on RWVJ (F_1.00_ = 17.894, *p* = 0.001, pEta^2^ = 0.249). However, the interaction effect of time and intervention (Time × Group) was not significant (F_5.00_ = 3.508, *p* = 0.318, pEta^2^ = 0.101). Bonferroni test results showed significant increases in RWVJ in the OMEGA + USRPT (*p* = 0.030), PRO + USRPT (*p* = 0.011), and PRO + OMEGA + USRPT (*p* = 0.008) groups in the post-test compared to the pre-test. However, no significant increase in RWVJ was observed in the USRPT, PLA + USRPT, or CON groups (*p* > 0.05). Additionally, no significant differences were observed between the research groups in the pre-test and post-test RWVJ records (*p* > 0.05; Figure 4, Table 4).

Sprint Index (SI): The results of the repeated measure analysis test showed that the main effect of the intervention was significant on SI (F_1.00_ = 17.358, *p* = 0.003, pEta^2^ = 0.157). Also, the interaction effect of time and intervention (Time × Group) was significant (F_5.00_ = 1.449, *p* = 0.013, pEta^2^ = 0.139). The Bonferroni test results indicated significant increases in SI in the OMEG.SRPT (*p* = 0.011) and PRO + OMEGA + USRPT (*p* = 0.004) groups in the post-test compared to the pre-test. However, no significant improvement in SI was observed in the PRO + USRPT, USRPT, PLA + USRPT, or CON groups (*p* > 0.05). Additionally, no significant differences were observed between the research groups in the pre-test and post-test SI records (*p* > 0.05; Figure 4, Table 4).

Velocity (V): The results of the repeated measure analysis test showed that the main effect of the intervention was significant on V (F_1.00_ = 10.092, *p* = 0.002, pEta^2^ = 0.157). Additionally, the interaction effect of time and intervention (Time × Group) was significant (F_5.00_ = 1.107, *p* = 0.031, pEta^2^ = 0.129). The Bonferroni test results indicated significant increases in V in the PRO + OMEGA + USRPT group (*p* = 0.003) in the post-test compared to the pre-test. However, no significant improvement in V was observed in the OMEGA + USRPT, PRO + USRPT, USRPT, PLA + USRPT, or CON groups (*p* > 0.05). Additionally, no significant differences were found between the research groups in the pre-test and post-test V records (*p* > 0.05; Figure 4, Table 4).

Fatigue Index (FI): The results of the repeated measure analysis test showed that the main effect of the intervention (F_1.00_ = 4.450, *p* = 0.437, pEta^2^ = 0.076) and the interaction effect of time and intervention (Time × Group) (F_5.00_ = 0.541, *p* = 0.744, pEta^2^ = 0.048) was not significant. Additionally, there were no significant differences between the research groups in the pre-test and post-test FI records (*p* > 0.05; Figure 4, Table 4).

Push-Off Distance (PD): The results of the repeated measure analysis test showed that the main effect of the intervention was significant on PD (F_1.00_ = 4.370, *p* = 0.003, pEta^2^ = 0.546). However, the interaction effect of time and intervention (Time × Group) was not significant (F_5.00_ = 1.759, *p* = 0.137, pEta^2^ = 0.140). The Bonferroni test results showed significant increases in PD in the OMEGA + USRPT (*p* = 0.001), PRO + USRPT (*p* = 0.020), and PRO + OMEGA + USRPT (*p* = 0.001) groups in the post-test compared to the pre-test. However, no significant increase was observed in PD in the USRPT, PLA + USRPT, or CON groups (*p* > 0.05). Additionally, there were no significant differences between the research groups in the pre-test and post-test PD records (*p* > 0.05; Figure 4, Table 4).

### 3.2. Dry-Land Performance Metrics

Vertical Jump Height (VJH): The results of the repeated measure analysis test showed that the main effect of the intervention was significant on VJH (F_1.00_ = 51.225, *p* = 0.001, pEta^2^ = 0.487). However, the interaction effect of time and intervention (Time × Group) was not significant (F_5.00_ = 1.836, *p* = 0.121, pEta^2^ = 0.145). The Bonferroni test results indicated significant increases in VJH in PLA + USRPT (*p* = 0.033), OMEGA + USRPT (*p* = 0.001), PRO + USRPT (*p* = 0.001), and PRO + OMEGA + USRPT (*p* = 0.001) groups in the post-test compared to the pre-test. However, no significant improvement in VJH was observed in the USRPT or CON groups (*p* > 0.05). Additionally, no significant differences were found between the research groups in the pre-test and post-test VJH records (*p* > 0.05; Figure 5, Table 4).

Repeated Vertical Jumps (RVJs): The results of the repeated measure analysis test showed that the main effect of the intervention was significant on RVJ (F_1.00_ = 22.144, *p* = 0.001, pEta^2^ = 0.297). However, the interaction effect of time and intervention (Time × Group) was not considerable (F_5.00_ = 0.790, *p* = 0.562, pEta^2^ = 0.068). The Bonferroni test results indicated significant increases in RVJ in PRO + USRPT (*p* = 0.015) and PRO + OMEGA + USRPT (*p* = 0.002) groups in the post-test compared to the pre-test. However, no significant improvement in RVJ was observed in OMEGA + USRPT, USRPT, PLA + USRPT, or CON groups (*p* > 0.05). Additionally, there were no significant differences between the research groups in the pre-test and post-test RVJ records (*p* > 0.05; Figure 5, Table 4).

Reaction Time (RT): The results of the repeated measure analysis test showed that the main effect of the intervention was significant on RT (F_1.00_ = 15.748, *p* = 0.001, pEta^2^ = 0.226). Moreover, the interaction effect of time and intervention (Time × Group) was significant (F_5.00_ = 3.431, *p* = 0.009, pEta^2^ = 0.241). The Bonferroni test results indicated significant improvements in RT in the PRO + OMEGA + USRPT group only (*p* = 0.001) in the post-test compared to the pre-test. No significant improvement in RT was observed in the OMEGA + USRPT, PRO + USRPT, USRPT, PLA + USRPT, or CON groups (*p* > 0.05). Additionally, there were no significant differences between the research groups in the pre-test and post-test RT records (*p* > 0.05; Figure 5, Table 4).

Agility: The results of the repeated measure analysis test showed that the main effect of the intervention was significant on agility (F_1.00_ = 26.818, *p* = 0.001, pEta^2^ = 0.336). Moreover, the interaction effect of time and intervention (Time × Group) was significant (F_5.00_ = 1.015, *p* = 0.026, pEta^2^ = 0.216). The Bonferroni test results indicated significant improvements in agility scores in OMEGA + USRPT (*p* = 0.010), PRO + USRPT (*p* = 0.019), and PRO + OMEGA + USRPT (*p* = 0.001) groups in the post-test compared to the pre-test. No significant improvement in agility was observed in the USRPT, PLA + USRPT, or CON groups (*p* > 0.05). Additionally, no significant differences were found between the research groups in the pre-test and post-test agility scores (*p* > 0.05; Figure 5, Table 4).

Fat Percentage (FP): The results of the repeated measure analysis test showed that the main effect of the intervention was significant on FP (F_1.00_ = 27.855, *p* = 0.001, pEta^2^ = 0.429). However, the interaction effect of time and intervention (Time × Group) was not significant (F_5.00_ = 1.890, *p* = 0.111, pEta^2^ = 0.149). The Bonferroni test results indicated significant reductions in FP in PRO + USRPT (*p* = 0.001) and PRO + OMEGA + USRPT (*p* = 0.001) groups in the post-test compared to the pre-test. However, no significant changes in FP were observed in OMEGA + USRPT, USRPT, PLA + USRPT, or CON groups (*p* > 0.05). Additionally, no significant differences were found between the research groups in the pre-test and post-test FP records (*p* > 0.05; Figure 5, Table 4).

Muscle Percentage (MP): The results of the repeated measure analysis test showed that the main effect of the intervention was significant on MP (F_1.00_ = 11.039, *p* = 0.002, pEta^2^ = 0.170). However, the interaction effect of time and intervention (Time × Group) was not significant (F_5.00_ = 0.605, *p* = 0.697, pEta^2^ = 0.053). The Bonferroni test results indicated significant increases in MP in the PRO + OMEGA + USRPT group only (*p* = 0.008) in the post-test compared to the pre-test. No significant improvement in MP was observed in the OMEGA + USRPT, PRO + USRPT, USRPT, PLA + USRPT, or CON groups (*p* > 0.05). Additionally, no significant differences were found between the research groups in the pre-test and post-test MP records (*p* > 0.05; Figure 5, Table 4).

Medicine Ball Throw (TM): The repeated measure analysis test results showed that the intervention’s main effect (F_1.00_ = 4.980, *p* = 0.273, pEta^2^ = 0.084), and the interaction effect of time and intervention (Time × Group) (F_5.00_ = 0.632, *p* = 0.697, pEta^2^ = 0.055) was insignificant. Additionally, no significant differences were observed between the research groups in the pre-test and post-test TM records (*p* > 0.05; Figure 4, Table 4).

## 4. Discussion

### 4.1. Summary of Key Findings

This study aimed to investigate the effects of probiotics or omega-3 fatty acid supplementation, either alone or in combination, in conjunction with USRPT, on specific performance indicators in sprint swimmers. The findings demonstrated that this combined intervention significantly enhanced both in-water and dry-land performance metrics among competitive swimmers. These improvements were particularly evident in sprint performance, anaerobic power, and agility metrics. The results highlight the synergistic effects of targeted supplementation and high-intensity interval training, offering valuable insights and practical applications for optimizing training adaptations, recovery, and competitive outcomes in aquatic sports. Supplementation with a symbiotic, the combination of a probiotic with an omega-3 fatty acid prebiotic, had a more pronounced impact on performance indicators compared to the individual pro- and prebiotic ingredients.

### 4.2. Potential Mechanisms of Probiotic Effects

The multi-strain probiotic blend was selected for its potential to improve gut health and immunity, which are relevant to recovery and adaptation in athletes. For example, *Lactiplantibacillus plantarum* BP06 has demonstrated anti-inflammatory properties and improved gut permeability, facilitating nutrient absorption and recovery from exercise-induced stress [12,49]. Similarly, *Bifidobacterium longum* BL03 has been shown to modulate immune responses and reduce gastrointestinal discomfort during exercise, which is critical for maintaining performance in endurance sports [50,51]. Strains such as *Lactobacillus acidophilus* BA05 and *Bifidobacterium breve* BB02 have been associated with reducing oxidative stress and enhancing immune function, helping to minimize fatigue and support recovery in physically active populations [52,53]. Additionally, *Lactobacillus bulgaricus* BD08 and *Streptococcus thermophilus* BT01, commonly used in multi-strain blends, contribute to the stabilization of the gut microbiota and improved protein metabolism, which are essential for recovery and adaptation in high-intensity training regimens [54]. These findings provide a mechanistic explanation for the observed improvements in anaerobic capacity and reduced fatigue in this study. While the effectiveness of this specific blend is supported by its components, further research is needed to explore its long-term benefits and optimize its application in sports. Future studies should investigate the interactions between these strains to understand better their combined effects on endurance, recovery, and overall performance in athletic populations.

In our study, the combination of probiotics and omega-3 supplementation with USRPT led to significant reductions in 50 m (−1.92%, ~0.50 s, *p* = 0.002, pEta^2^ = 0.286) and 100 m freestyle times (−2.48%, ~0.73 s, *p* = 0.041, pEta^2^ = 0.229), alongside improvements in sprint index and vertical jump performance, with the partial eta-squared values indicating moderate to large effects. These improvements, although numerically modest, are practically meaningful in sprint swimming, where even differences of 0.1–0.3 s often determine medal rankings. This aligns in part with Lee et al. (2024), who demonstrated that six weeks of exercise combined with *Lactiplantibacillus plantarum* PL-02 and *Lactococcus lactis* LY-66 strains enhanced muscle strength, power, and endurance, supported by shifts in the gut microbiota and reduced inflammatory markers [55]. Similarly, Imanian et al. (2024) found that probiotics with casein modestly improved aerobic capacity in male soccer players, particularly when consumed together, indicating a possible synergistic effect [12]. In contrast, other studies report limited ergogenic effects: Lamprecht et al. (2012) observed no changes in VO_2_ max or maximal performance following multi-strain probiotic use in endurance-trained men [56], while Pugh et al. (2020) found minor metabolic shifts without time-trial improvements in cyclists after probiotic supplementation [57]. Notably, Carbuhn et al. (2018) examined Bifidobacterium longum in swimmers and found no significant effects on anaerobic or vertical jump outcomes [58]. Such discrepancies may stem from differences in strain diversity, dosages, intervention durations, or sport-specific demands. Importantly, given typical handheld timing variability (±0.24 s over 50 m), the observed changes in our study exceed expected measurement error, reinforcing that these statistically significant improvements likely reflect true physiological adaptations.

In the present study, the combined supplementation of probiotics and omega-3 with USRPT resulted in significant improvements in key performance metrics, including sprint index, vertical jump height, and reaction time. These changes were accompanied by the partial eta-squared values indicating moderate to large effects, suggesting physiologically meaningful adaptations that exceed typical measurement error margins. Mechanistically, probiotics may contribute by modulating intestinal permeability and enhancing nutrient absorption [59], while also reducing inflammation and promoting villus development. Their role in bolstering innate and adaptive immunity—through increased immunoglobulin production, heightened antimicrobial activity, and improved T- and B-cell function—further supports recovery and resilience during demanding protocols such as USRPT. These proposed mechanisms align with earlier findings: Salarkia et al. (2013) reported that probiotic yogurt reduced respiratory infections and supported VO_2_ max in endurance swimmers [60], while collegiate female swimmers supplemented with probiotics showed improved recovery-stress scores [58]. Such evidence underscores probiotics’ potential to mitigate physiological stress and accelerate recovery [61]. Importantly, our findings expand on this by demonstrating that the PRO + OMEGA + USRPT group experienced the most pronounced improvements, pointing toward a synergistic effect of probiotics and omega-3 on anaerobic performance and energy metabolism. This highlights the value of integrated nutritional and training strategies tailored specifically to the demands of sprint swimming.

### 4.3. Body Composition and Probiotics’ Role

In line with the body composition findings of elite Iranian swimmers, Samanipour et al. (2024) reported that swimmers had a significantly lower body fat percentage and higher muscle mass percentage compared to athletes in other sports [3]. Specifically, the study found that swimmers exhibited a lower percentage of body fat and a higher percentage of fat-free mass, including skeletal muscle mass, which is crucial for swimming performance. These characteristics contribute to enhanced propulsion, endurance, and overall swimming efficiency. Samanipour et al. (2024) [3] observed that body composition profiles align with the positive effects observed in the present study, where probiotics may have supported muscle recovery and lean mass maintenance. In our study, the PRO + OMEGA + USRPT group demonstrated significant improvements in both muscle mass and a reduction in body fat percentage, which likely contributed to performance gains in swimming. Additionally, Samanipour et al. (2024) emphasized the importance of tailored nutritional strategies, particularly for optimizing fat and muscle mass to improve swimming efficiency [3]. This further supports the potential role of probiotics in managing body composition by helping maintain or increase lean muscle mass while potentially reducing fat, which is crucial for maximizing swimming performance. In summary, probiotics play a multifaceted role in enhancing swimmer performance, particularly when integrated with targeted training and nutritional strategies. The body composition and performance improvements observed in our study, along with supporting evidence from the literature, highlight the beneficial role of probiotics in optimizing muscle mass and promoting recovery. Future research should explore the long-term effects of probiotic supplementation, dose–response relationships, and interactions with other ergogenic aids to optimize athletic performance in aquatic sports.

### 4.4. Potential Mechanisms of Omega-3 Effects

In addition to probiotics, omega-3 fatty acids appear to play a meaningful role in enhancing performance adaptations among swimmers. In our study, the combined PRO + OMEGA + USRPT intervention resulted in significant improvements in sprint index, vertical jump height, and reaction time, with the partial eta-squared values indicating moderate to large effects that exceed typical handheld timing variability (approximately ±0.24 s over 50 m). Mechanistically, omega-3s may support muscle protein synthesis and strength by enhancing amino acid absorption and activating anabolic pathways such as AKT and mTOR [62], processes critical for generating propulsion in the water. They also mitigate muscle atrophy by downregulating catabolic markers, such as MAFbx, MURF, NF-κB, and ROS, potentially explaining the improved recovery metrics observed here [63]. Notably, while Guzmán et al. (2011) reported enhanced reaction speed and anaerobic performance with DHA-rich fish oil in soccer players [64], Lewis et al. (2017) found no interval speed improvements in active men after short-term omega-3 use [65], underscoring that sport type, duration, and baseline status may influence outcomes. Furthermore, omega-3s upregulate PGC-1, supporting mitochondrial biogenesis and angiogenesis, which aids both aerobic and anaerobic demands during swimming [66]. Although some studies suggest omega-3s may positively shape the gut microbiota and thereby complement probiotic actions [67], direct evidence for enhanced probiotic absorption remains limited. Still, our findings suggest that combined omega-3 and probiotic intake may offer synergistic benefits, as evidenced by the enhanced performance and body composition improvements in the PRO + OMEGA + USRPT group. However, it is important to note that the changes in fatigue index did not reach statistical significance, suggesting that anti-fatigue adaptations may be less robust or require longer interventions to emerge. Collectively, these results support a multifactorial strategy targeting both muscle and gut health to enhance swimmer performance. Future studies should examine longer durations, biochemical markers, and the sustainability of such adaptations to delineate these interactive effects fully.

### 4.5. Synergistic Effects of Combined Supplementation

USRPT has been identified as an effective method for enhancing sprint swimmers’ performance by improving key physiological and functional parameters. In the present study, USRPT significantly reduced heart rate, blood lactate, and perceived exertion, enhancing cardiovascular efficiency and reducing anaerobic stress during training sessions [23,25]. Additionally, USRPT improved swimming speed, maximal oxygen uptake (VO_2_ max), and anaerobic capacity, all of which are critical for sprint swimming performance. Enhanced strength and biomechanical efficiency further improved sprint-specific metrics, such as reduced 50 m and 100 m freestyle times. These results support the growing evidence that USRPT is an optimal training modality for improving neuromuscular adaptations and lactate clearance, which is critical for high-intensity swimming efforts [24]. Over time, the observed enhancements in muscle strength and maximal oxygen uptake suggest physiological adaptations, such as improved muscle fiber recruitment and mitochondrial efficiency, contributing to the swimmers’ overall enhanced performance.

The findings of this study align with prior research, reinforcing the effectiveness of USRPT in improving both short-term and long-term swimming efficiency. The improvements in anaerobic capacity, as reflected in metrics such as the sprint index and reduced fatigue index, further underscore the effectiveness of USRPT in preparing athletes for the intense physical demands of sprint events [25]. These findings are consistent with the hypothesis that repeated sprints at race pace promote sustained neuromuscular adaptations and improve overall energy utilization [23]. Furthermore, the study demonstrated that combining omega-3 and probiotics supplementation with USRPT yielded the most pronounced improvements in swimming performance metrics, as reflected by significant Time × Group interactions for primary outcomes, supporting a synergistic effect despite some overlapping confidence intervals. Omega-3 supplementation reduces exercise-induced muscle damage and enhances recovery through its anti-inflammatory and mitochondrial-supportive effects [14]. These properties were evident in improved metrics such as vertical jump height and reaction time. Meanwhile, probiotics supplementation supported gut health, nutrient absorption, and immune function, facilitating improved recovery and sustained high-intensity training performance [8]. Together, omega-3 and probiotics supplementation appeared to have synergistic effects, as omega-3 enhanced gut microbiota diversity, amplifying the benefits of probiotics on systemic inflammation and recovery [67]. Importantly, beyond these physiological mechanisms, our study demonstrated significant Time × Group interaction effects for primary outcomes such as 50m freestyle (F_5,54_ = 4.328, *p* = 0.002) and sprint index (F_5,54_ = 1.449, *p* = 0.013). These interactions indicate that the pattern of change over time differed substantially between intervention groups, providing formal statistical evidence that the combined intervention elicited adaptations beyond those achieved by individual treatments, consistent with an actual synergistic effect. This extends previous literature by explicitly showing that integrating nutritional strategies with targeted training can produce interactive benefits greater than additive effects alone, likely through complementary pathways involving gut integrity, anti-inflammatory responses, and optimized energy metabolism.

The combined intervention of omega-3 and probiotics supplementation with USRPT significantly reduced sprint times and enhanced anaerobic power, highlighting the value of integrating targeted training with strategic nutritional support to maximize swimming performance. In our study, this approach resulted in a 1.92% reduction in 50 m freestyle time (approximately 0.48 s) and a 2.48% reduction in 100 m time (around 1.33 s). Although these improvements may appear modest, in competitive sprint swimming, even small gains can be decisive, often determining medal positions, as differences of just 0.1–0.3 s frequently separate podium finishes in elite events [68]. This underscores the well-established principle that among highly trained athletes, where physiological ceilings tightly limit significant adaptations, incremental enhancements can substantially influence race outcomes [69]. These findings support the adoption of combined nutritional and training strategies to secure such critical performance advantages. Future research should investigate the long-term sustainability and dose–response aspects of these interventions to clarify their role in competitive swimming further.

### 4.6. Limitations and Directions for Future Research

This study has several limitations that should be acknowledged to properly frame the findings. It involved only male sprint swimmers with substantial competitive experience, which limits generalizability to female athletes or other sports. Although probiotics and omega-3s are known to influence the gut microbiota, we did not directly assess microbial shifts, leaving the mechanistic link to performance speculative. Similarly, the absence of physiological markers, such as lactate, inflammatory cytokines, or indicators of muscle damage, constrains our understanding of the processes driving these adaptations. Due to financial and laboratory constraints, we were also unable to characterize the probiotic strains and dosages precisely. While breakfast was standardized, broader dietary intake—especially habitual consumption of omega-3 fatty acids and fermented foods—was recorded but not strictly controlled, introducing possible confounding factors. The eight-week duration, though sufficient for detecting short-term changes, does not inform on longer-term sustainability. Importantly, while several outcomes showed moderate to large effects that exceed typical hand-timing variability (±0.24 s over 50 m), the fatigue index did not reach significance, suggesting that certain anti-fatigue benefits may be less robust or require longer interventions. These considerations underscore the need for future studies involving more diverse athlete cohorts, extended interventions, stricter dietary control, and comprehensive physiological and mechanistic assessments.

## 5. Conclusions

This study highlights the potential benefits of combining USRPT with probiotics and omega-3 supplementation to enhance sprint swimming performance. While USRPT alone improved anaerobic capacity, speed, and recovery, adding probiotics and omega-3 led to further gains, reflected in faster sprint times and better neuromuscular outcomes. Significant Time × Group interactions suggest possible synergistic effects, though the modest magnitude and overlapping confidence intervals advise cautious interpretation. These findings emphasize the importance of aligning training with nutrition to support optimal performance. Future studies should incorporate mechanistic endpoints (such as inflammatory or muscle-damage markers), longer follow-up periods, and the inclusion of female athletes to better understand the scope and sustainability of these adaptations.

## Figures and Tables

**Figure 1 nutrients-17-02296-f001:**
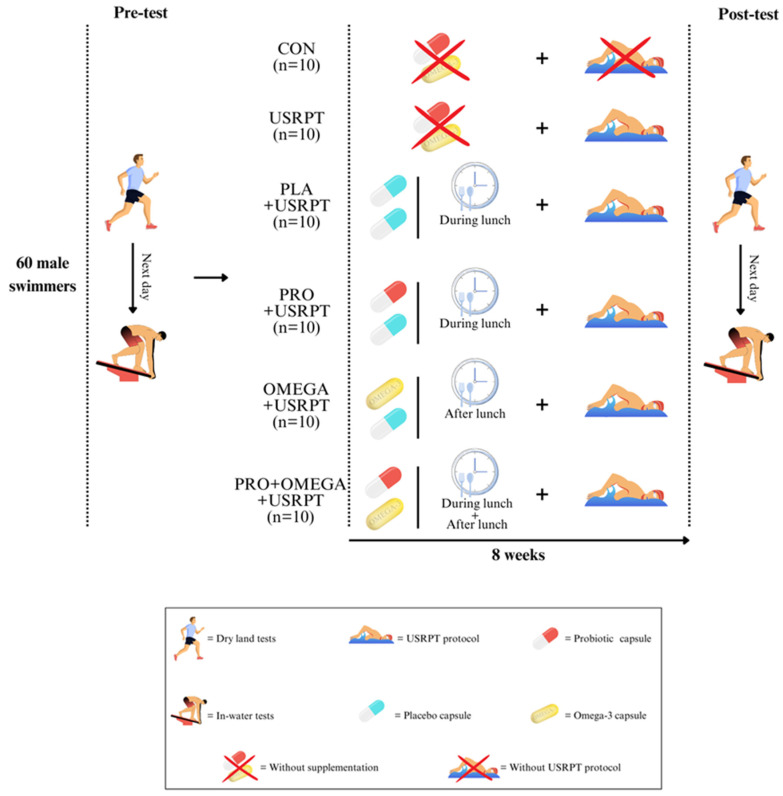
The protocol of the current study. Control (CON), training (USRPT), placebo and training (PLA + URSPT), probiotics and training (PRO + USRPT), omega-3 and training (OMEGA + USRPT), and probiotics with omega-3 and training (PRO + OMEGA + USRPT. All the capsules were identical in appearance and color; differences in the figure are only for better illustration. The figure was generated using CANVA (version 1.109.0, Graphic design Software, Sydney, Australia).

**Figure 2 nutrients-17-02296-f002:**
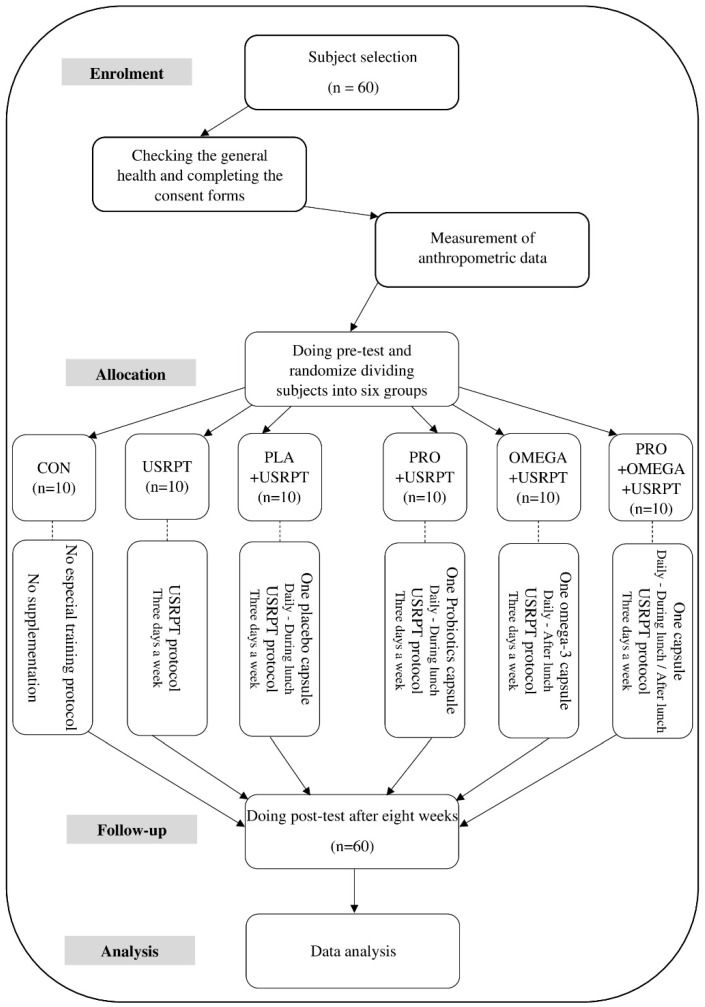
Flowchart illustrating the different phases of the research and study selection.

**Figure 3 nutrients-17-02296-f003:**
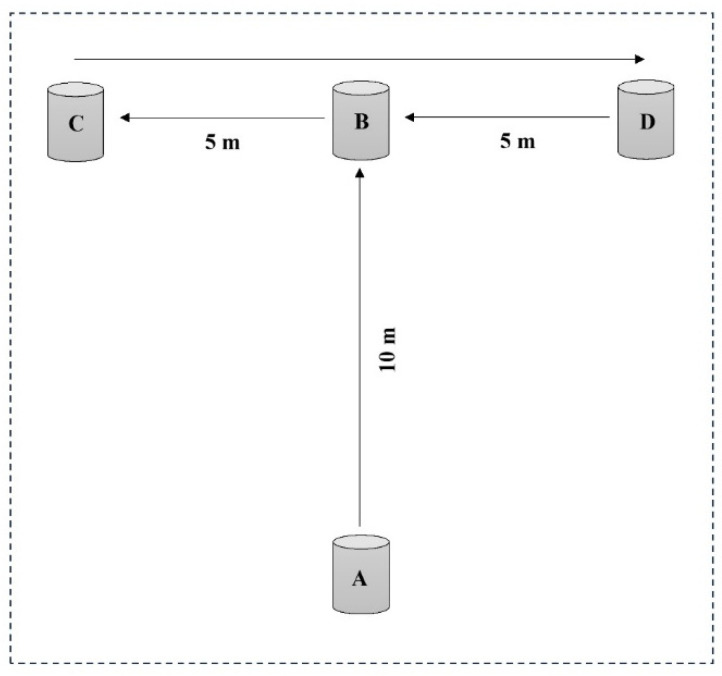
The agility *T*-test protocol.

**Figure 4 nutrients-17-02296-f004:**
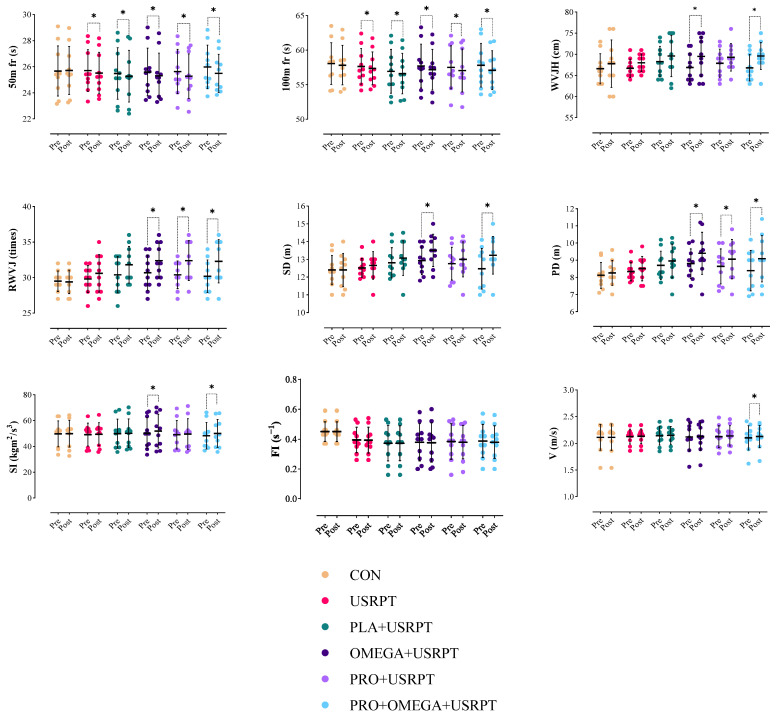
Means and standard deviations of the in-water test results in the six groups (control (CON), training (USRPT), placebo and training (PLA + URSPT), probiotics and training (PRO + USRPT), omega-3 and training (OMEGA + USRPT), and probiotics with omega-3 and training (PRO + OMEGA + USRPT)). Here, 50 m fr: 50 m freestyle; 100 m fr: 100 m freestyle; WVJH: vertical jump height in the pool or in-water vertical jump; RWVJ: repeated vertical jump in the pool; SD: start distance; PD: push-off distance in the water; SI: sprint index; FI: fatigue index; V: velocity. *: significant difference compared to the pre-test.

**Figure 5 nutrients-17-02296-f005:**
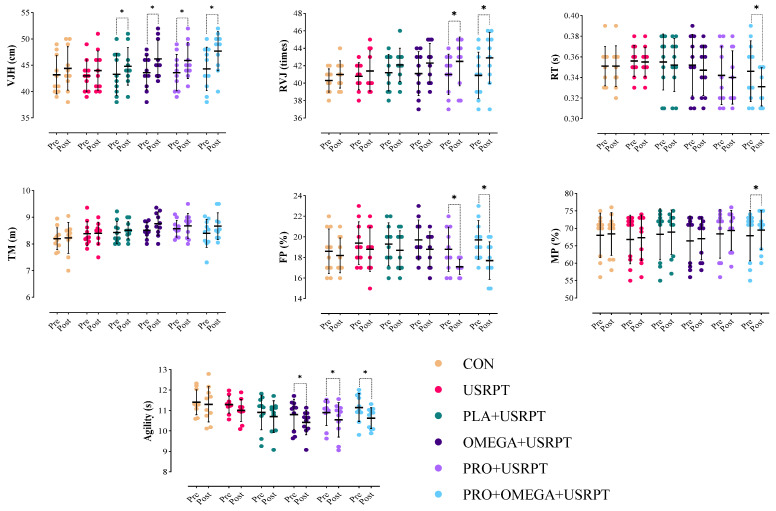
Means and standard deviations of the dry-land test results in the six groups (control (CON), training (USRPT), placebo and training (PLA + URSPT), probiotics and training (PRO + USRPT), omega-3 and training (OMEGA + USRPT), and probiotics with omega-3 and training (PRO + OMEGA + USRPT)). VJH: vertical jump height; RVJ: repeated vertical jump; RT: reaction time; TM: throwing a medicine ball; FP: body fat percentage; MP: muscle mass percentage. *: significant difference compared to the pre-test.

**Table 1 nutrients-17-02296-t001:** The anthropometric data of the participants.

Characteristic	Mean ± SD (n = 60)
Age (years)	23.20 ± 3.64
Height (cm)	182.20 ± 5.21
Weight (kg)	81.6 ± 4.42

**Table 2 nutrients-17-02296-t002:** Strains and dosage per one capsule (200 mg) of the probiotics used in the present study.

Strains	Dosage (CFU)
*Lactiplantibacillus plantarum* BP06	0.43 × 10^11^
*Lacticaseibacillus casei* BP07	0.65 × 10^11^
*Lactobacillus acidophilus* BA05	0.94 × 10^11^
*Lactobacillus bulgaricus* BD08	0.36 × 10^11^
*Bifidobacterium infantis* BI04	0.57 × 10^11^
*Bifidobacterium longum* BL03	0.73 × 10^11^
*Bifidobacterium breve* BB02	0.62 × 10^11^
*Streptococcus thermophilus* BT01	0.20 × 10^11^
Total	4.5 × 10^11^

**Table 3 nutrients-17-02296-t003:** Means and standard deviation (SD) of the pre-test and post-test measured variables (n = 10 in each group).

				CON	USRPT	PLA + USRPT	OMEGA + USRPT	PRO + USRPT	PRO + OMEGA + USRPT
In-water tests	50 m fr (s)	Mean	Pre	25.66	25.70	25.48	25.59	25.63	25.99
		SD		1.91	1.59	1.97	1.83	1.70	1.66
		Mean	Post	25.71	25.51	25.27	25.30	25.28	25.49
		SD		1.83	1.59	1.98	1.73	1.70	1.45
	100 m fr (s)	Mean	Pre	58.05	57.64	56.93	57.70	57.51	57.82
		SD		3.02	2.58	3.17	3.16	3.12	3.15
		Mean	Post	57.82	57.34	56.60	57.17	57.05	57.09
		SD		2.86	2.38	2.93	2.93	3.12	2.80
	WVJH (cm)	Mean	Pre	66.60	66.70	68.30	66.90	67.90	66.80
		SD		3.56	2.16	3.62	3.44	3.34	3.15
		Mean	Post	67.80	68.00	69.60	69.50	69.30	69.60
		SD		5.61	2.05	4.88	4.71	3.26	3.20
	RWVJ (times)	Mean	Pre	29.50	29.80	30.40	30.70	30.40	30.20
		SD		1.43	1.87	2.54	2.49	1.95	2.29
		Mean	Post	29.40	30.60	31.80	32.40	32.40	32.30
		SD		1.64	2.63	2.57	2.45	2.79	3.05
	SD (m)	Mean	Pre	12.40	12.52	12.81	12.93	12.76	12.47
		SD		0.82	0.53	0.85	0.78	0.92	1.14
		Mean	Post	12.40	12.66	13.07	13.51	13.00	13.23
		SD		0.94	0.77	0.97	0.90	0.99	1.06
	PD (m)	Mean	Pre	8.12	8.34	8.70	8.80	8.64	8.39
		SD		0.75	0.54	0.82	0.86	1.01	1.17
		Mean	Post	8.26	8.51	8.96	9.40	9.06	9.09
		SD		0.72	0.71	0.98	1.22	1.15	1.35
	SI (kgm^2^/s^3^)	Mean	Pre	49.71	49.13	49.84	50.27	49.03	48.24
		SD		10.05	8.77	11.20	11.98	11.01	10.16
		Mean	Post	49.83	49.42	50.17	51.82	49.55	50.00
		SD		10.33	8.83	11.07	13.07	11.86	10.76
	FI (s^–1^)	Mean	Pre	0.44	0.39	0.37	0.37	0.38	0.38
		SD		0.06	0.08	0.11	0.12	0.12	0.11
		Mean	Post	0.44	0.39	0.37	0.37	0.37	0.37
		SD		0.06	0.08	0.11	0.13	0.11	0.10
	V (m/s)	Mean	Pre	2.11	2.12	2.14	2.12	2.12	2.10
		SD		0.23	0.13	0.17	0.26	0.20	0.22
		Mean	Post	2.11	2.13	2.15	2.12	2.13	2.12
		SD		0.24	0.13	0.16	0.25	0.19	0.20
Dry land tests	VJH (cm)	Mean	Pre	43.20	43.00	43.30	43.60	43.60	44.30
		SD		3.64	3.01	4.00	3.23	3.43	4.11
		Mean	Post	44.40	44.00	44.80	46.20	45.90	47.70
		SD		4.14	3.71	3.55	3.55	3.41	3.71
	RVJ (times)	Mean	Pre	40.30	40.80	41.20	41.10	41.00	40.90
		SD		1.33	1.54	2.09	2.51	2.30	2.64
		Mean	Post	41.00	41.40	42.10	42.30	42.50	42.90
		SD		1.56	2.31	1.91	2.26	2.79	2.92
	RT (s)	Mean	Pre	0.35	0.35	0.35	0.35	0.34	0.34
		SD		0.01	0.01	0.02	0.02	0.02	0.03
		Mean	Post	0.35	0.35	0.35	0.34	0.34	0.33
		SD		0.01	0.01	0.02	0.02	0.02	0.01
	TM (m)	Mean	Pre	8.19	8.38	8.42	8.51	8.56	8.39
		SD		0.41	0.45	0.40	0.30	0.31	0.52
		Mean	Post	8.22	8.40	8.51	8.75	8.67	8.66
		SD		0.57	0.41	0.32	0.43	0.46	0.50
	FP (%)	Mean	Pre	18.60	19.40	19.30	19.70	18.80	19.70
		SD		2.17	2.06	2.05	1.94	2.14	1.88
		Mean	Post	18.20	18.80	18.70	18.80	17.10	17.70
		SD		1.68	2.14	1.88	1.68	0.73	1.82
	MP (%)	Mean	Pre	68.00	66.80	68.30	66.40	68.40	67.90
		SD		6.32	6.98	7.30	7.12	6.93	7.21
		Mean	Post	68.40	67.30	68.90	67.00	69.40	69.50
		SD		6.13	6.53	6.38	6.01	5.73	5.56
	Agility (s)	Mean	Pre	11.40	11.28	10.90	10.80	10.89	11.14
		SD		0.60	0.41	0.84	0.74	0.64	0.68
		Mean	Post	11.30	11.00	10.70	10.41	10.54	10.60
		SD		0.86	0.53	0.77	0.58	0.83	0.51

CON: control; USRPT: training; PLA + URSPT: placebo and training; PRO + USRPT: probiotics and training; OMEGA + USRPT: omega-3 and training; PRO + OMEGA + USRPT: probiotics with omega-3 and training; 50 m fr: 50 m freestyle; 100 m fr: 100 m freestyle; WVJH: vertical jump height in the pool or in-water vertical jump; RWVJ: repeated vertical jump in the pool; SD: start distance; PD: push-off distance in the water; SI: sprint index; FI: fatigue index; V: velocity; VJH: vertical jump height; RVJ: repeated vertical jump; RT: reaction time; TM: throwing a medicine ball; FP: body fat percentage; MP: muscle mass percentage; s: second; cm: centimeter; m: meter; kg: kilogram.

**Table 4 nutrients-17-02296-t004:** Comparison of the variable data between pre-test and post-test in the six groups.

Variables				CON	USRPT	PLA + USRPT	OMEGA + USRPT	PRO + USRPT	PRO + OMEGA + USRPT
				Post					
In-water tests	50 m fr (s)	MD	Pre	0.04	−0.19	−0.21	−0.28	−0.35	−0.50
		Sig		0.587	0.033	0.017	0.002	0.001	0.001
		95% CI		−0.12–0.22	−0.36–−0.01	−0.39–−0.04	−0.46–−0.11	−0.52–−0.17	−0.67–−0.32
	100 m fr (s)	MD	Pre	−0.23	−0.29	−0.33	−0.53	−0.45	−0.72
		Sig		0.104	0.043	0.024	0.001	0.002	0.001
		95% CI		−0.52–0.05	−0.58–−0.01	−0.61–−0.04	−0.81–−0.24	−0.73–−0.16	−1.01–−0.43
	WVJH (cm)	MD	Pre	1.20	1.30	1.30	2.60	1.40	2.80
		Sig		0.141	0.111	0.111	0.002	0.087	0.001
		95% CI		−0.41–2.81	−0.31–2.91	−0.31–2.91	0.99–4.21	−0.21–3.01	1.19–4.41
	RWVJ (times)	MD	Pre	−0.10	0.80	1.40	1.70	2.00	2.10
		Sig		0.896	0.299	0.72	0.030	0.011	0.008
		95% CI		−1.62–1.42	−0.72–2.32	−0.12–2.92	0.17–3.22	0.47–3.52	0.57–3.62
	SD (m)	MD	Pre	0.00	0.14	0.26	0.58	0.24	0.76
		Sig		1.000	0.466	0.178	0.004	0.213	0.001
		95% CI		−0.38–0.38	−0.24–0.52	−0.12–0.64	0.19–0.96	−0.14–0.62	0.37–1.14
	PD (m)	MD	Pre	0.14	0.17	0.26	0.60	0.42	0.70
		Sig		0.426	0.335	0.143	0.001	0.020	0.001
		95% CI		−0.21–0.49	−0.18–0.52	−0.09–0.61	0.25–0.95	0.07–0.77	0.35–1.05
	SI (kgm^2^/s^3^)	MD	Pre	0.11	0.29	0.33	1.55	0.51	1.76
		Sig		0.851	0.616	0.576	0.011	0.386	0.004
		95% CI		−1.06–1.29	−0.88–1.47	−0.84–1.51	0.37–2.72	−0.66–1.69	0.58–2.94
	FI (s^−1^)	MD	Pre	0.000	−0.001	−0.001	−0.003	−0.005	−0.005
		Sig		0.891	0.837	0.681	0.356	0.097	0.079
		95% CI		−0.006–0.005	−0.006–0.005	−0.007–0.005	−0.009–0.003	−0.011–0.001	−0.011–0.001
	V (m/s)	MD	Pre	0.001	0.005	0.008	0.009	0.014	0.025
		Sig		0.901	0.533	0.320	0.264	0.085	0.003
		95% CI		−0.015–0.017	−0.011–0.021	−0.008–0.024	−0.007–0.025	−0.002–0.030	0.009–0.041
Dry land tests	VJH (cm)	MD	Pre	1.20	1.00	1.50	2.60	2.30	3.40
		Sig		0.085	0.150	0.033	0.001	0.001	0.001
		95% CI		−0.17–2.57	−0.37–2.37	0.12–2.87	1.22–3.97	0.92–3.67	2.02–4.77
	RVJ (times)	MD	Pre	0.70	0.60	0.90	1.20	1.50	2.00
		Sig		0.247	0.321	0.139	0.061	0.015	0.002
		95% CI		−0.50–1.90	−0.60–1.80	−0.30–2.10	−0.10–2.40	0.30–2.70	0.80–3.20
	RT (s)	MD	Pre	−0.001	−0.002	−0.001	−0.004	−0.002	−0.013
		Sig		0.582	0.367	0.637	0.095	0.433	0.001
		95% CI		−0.006–0.004	−0.007–0.003	−0.006–0.004	−0.009–0.001	−0.007–0.003	−0.018–0.008
	TM (m)	MD	Pre	0.02	0.01	0.08	0.24	0.11	0.27
		Sig		0.850	0.919	0.541	0.083	0.424	0.061
		95% CI		−0.24–0.30	−0.26–0.28	−0.19–0.35	−0.03–0.51	−0.16–0.38	−0.01–0.54
	FP (%)	MD	Pre	−0.40	−0.60	−0.60	−0.90	−1.70	−2.00
		Sig		0.408	0.216	0.216	0.066	0.001	0.001
		95% CI		−1.36–0.56	−1.56–0.36	−1.56–0.36	−1.86–0.06	−2.66–−0.73	−2.96–−1.03
	MP (%)	MD	Pre	0.40	0.50	0.60	0.60	1.00	1.60
		Sig		0.492	0.390	0.303	0.303	0.089	0.008
		95% CI		−0.75–1.55	−0.65–1.65	−0.55–1.75	−0.55–1.75	−0.15–2.15	0.44–2.75
	Agility (s)	MD	Pre	−0.10	−0.28	−0.20	−0.39	−0.35	−0.52
		Sig		0.488	0.066	0.172	0.010	0.019	0.001
		95% CI		−0.39–0.19	−0.57–0.01	−0.49–0.09	−0.68–−0.09	−0.64–−0.05	−0.81–−0.22

CON: control; USRPT: training; PLA + URSPT: placebo and training; PRO + USRPT: probiotics and training; OMEGA + USRPT: omega-3 and training; PRO + OMEGA + USRPT: probiotics with omega-3 and training; 50 m fr: 50 m freestyle; 100 m fr: 100 m freestyle; WVJH: vertical jump height in the pool or in-water vertical jump; RWVJ: repeated vertical jump in the pool; SD: start distance; PD: push-off distance in the water; SI: sprint index; FI: fatigue index; V: velocity; VJH: vertical jump height; RVJ: repeated vertical jump; RT: reaction time; TM: throwing a medicine ball; FP: body fat percentage; MP: muscle mass percentage; s: second; cm: centimeter; m: meter; kg: kilogram; MD: mean difference; CI: confidence interval.

## Data Availability

The original contributions presented in this study are included in the article. Further inquiries can be directed to the corresponding author.

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
