# Peer review of "Synergistic Effects of Probiotic and Omega-3 Supplementation with Ultra-Short Race Pace Training on Sprint Swimming Performance"

_nutrients, 2025, doi:10.3390/nu17142296_

Round 1

Reviewer 1 Report

Comments and Suggestions for Authors

Having read the manuscript attentively, I would first like to acknowledge the authors’ effort in bringing together supplementation, training methodology and performance testing in a single experiment. The idea of combining a multi-strain probiotic with omega-3 fatty acids while swimmers follow ultra-short race-pace training (USRPT) is certainly topical, and the paper sits well within the scope of Nutrients. That said, several aspects need careful refinement before the work can be considered for publication.

Beginning with the title and abstract, the study is correctly announced as examining “synergistic effects”, yet no statistical interaction term is reported to substantiate synergy. The abstract is informative but over-loaded with numeric detail; readers would be better served if the most salient findings—magnitude of change in sprint times and the corresponding effect sizes—were highlighted, leaving secondary variables for the main text. Minor style issues are present (for instance, “providing valuable” instead of “offer valuable” in the final sentence) .

The introduction is thorough but occasionally repetitive. Sections that review probiotic mechanisms, omega-3 actions and USRPT benefits overlap; a tighter narrative focusing on the knowledge gap (i.e., whether co-supplementation adds benefit to USRPT) would sharpen the rationale. Some statements, such as probiotics “strengthen the buffering system” or omega-3 acting as a prebiotic, are intriguing but speculative; they need citations of primary mechanistic work or should be phrased more cautiously . Finally, the introduction would profit from a short paragraph explaining why only male swimmers were recruited and acknowledging, up-front, the limited generalisability that follows.

The methods section is generally detailed, yet a few omissions compromise reproducibility. Six groups of ten participants are presented as adequately powered; however, the G*Power calculation quoted assumes an effect size of 0.25 and only two repeated measures, whereas the study analyses more than a dozen dependent variables, greatly inflating the family-wise Type-I error risk . A justification of the chosen alpha level or the use of an adjusted procedure (e.g., Holm or Benjamini–Hochberg) is required. Compliance with supplementation is not quantified, and the possibility that the fish-oil capsule’s odour unblinded participants is not discussed. Although breakfast was standardised, total dietary intake—particularly habitual omega-3 and fermented-food consumption—was merely “registered” and not controlled, which could confound outcomes.

With respect to statistical analysis, reliance on repeated-measures ANOVA plus Bonferroni tests is acceptable for two time points, yet the paper does not report whether sphericity or homogeneity of covariance was assessed. Partial eta-squared is presented but the thresholds adopted (0.01, 0.059, 0.138) are mistakenly attributed to Cohen; Cohen discussed f, not η². More importantly, what readers need is a clear group × time interaction term for each primary endpoint to verify the alleged synergy. Mixed-effects modelling would have offered a more flexible approach with unequal variances and missing data. Several p-values are given as “P = 0.000”; these should be stated as P < 0.001 following journal style.

Turning to the results, the authors provide large tables but very little narrative interpretation. For example, a 1.92 % reduction in 50 m time equates to roughly half a second at this swimming level and may or may not be practically meaningful; the discussion should relate the changes to race outcomes. In many variables, confidence intervals overlap widely across groups, so the claim that the combination group “exhibited the most substantial improvements across performance metrics” feels overstated . Graphs are included, yet some axes lack units (e.g., figure depicting vertical-jump counts) and the asterisks denoting significance do not correspond to specific comparisons, making the figures hard to read .

The discussion is extensive but drifts into a literature review that occasionally eclipses the study’s own findings. Parts summarise omega-3 research in soccer players and coronary-artery patients, material that, while interesting, distracts from the swimming context . The authors interpret every statistically significant p-value as physiologically important without reflecting on effect magnitude or measurement error (hand-held stopwatch times can vary by ±0.24 s over 50 m). Conversely, the non-significant fatigue-index data are not examined, yet might challenge the proposed anti-fatigue narrative. Limitations are acknowledged—particularly the absence of gut-microbiota data and the exclusive enrolment of men—but these appear late in the manuscript and should also mention the brief, eight-week duration and lack of biochemical markers of inflammation or muscle damage .

The conclusion repeats much of the discussion and would benefit from restraint: stating that the combination “significantly benefits” sprint swimming is accurate, but the magnitude and practical impact need moderation. Suggestions for future work should specify the need for mechanistic endpoints, longer follow-up and inclusion of female athletes.

Author Response

Dear reviewer, 

Reviewer 2 Report

Comments and Suggestions for Authors

This is a very well written MS on a very interesting topic.

Only three minor points:

  1. Section 2.3 : For how many days were the different supplements were provided to the athletes before assessing speed and the other parameters?
  2. Line 97: the commercially available omega-3s are ethyl esters and not phospholipids. This line needs editing to reflect this.
  3. Table 3 : it is not easy for the reader to follow. How many participants took part at each style?

I am happy to review the revised MS.

Author Response

Dear reviewer,

Reviewer 3 Report

Comments and Suggestions for Authors

Dear Authors,

I found your study both interesting and valuable.

However, I have a few minor comments for your consideration:

  • Please clearly describe the novelty of your study in the Introduction section.

  • Ensure that all references are formatted according to the MDPI guidelines.

  • Figure 1- include the name of the software used to create the figure.

  • Figure 2 is unclear and difficult to read. Consider revising or removing it.

  • I recommend adding a section discussing the limitations of the study.

  • The conclusion should be more comprehensive and in-depth.

Author Response

Dear reviewer,

Round 2

Reviewer 1 Report

Comments and Suggestions for Authors

The authors responded to all suggestions. The manuscript is suitable for pubblication

Author Response

Thank You.

Reviewer 2 Report

Comments and Suggestions for Authors

The MS can now be accepted.

Thank you to the authors.

Author Response

Thank you.